# Analysis of OSTBC-OFDM Combined with Dual-Polarization and Time-Diversity in Millimeter-Wave MIMO Channels with Rain Distortions

**DOI:** 10.3390/s22197182

**Published:** 2022-09-22

**Authors:** Avner Elgam, Yossi Peretz, Yosef Pinhasi

**Affiliations:** 1Faculty of Engineering, Ariel University, Ariel 40700, Israel; 2Department of Computer Sciences, Lev Academic Center, Jerusalem College of Technology, Jerusalem 9372115, Israel

**Keywords:** rain distortion, XPI mechanism, MIMO diversity techniques, mmWave wireless communication systems

## Abstract

Various destructive weather and physical phenomena affect many parameters in the radio layer (i.e., affecting the wireless paths Over-The-Air (OTA)) of many outdoor-to-outdoor wireless systems. These destructive effects create polarization torsion and rotation of the signals propagating in space and cause the scattering of wireless spatial paths. The direct meaning is a significant degradation in system performance, especially in the Quality-of-Service (QoS). Under these challenging scenarios, intelligent utilization of advanced Multiple-Input-Multiple-Output (MIMO) techniques such as polarization-diversity and time-diversity at the transmitter, as well as at the receiver, and intelligent use of the Cross-Polarization-Isolation (XPI) mechanism, are essential. We prove that combining these techniques with the tuning of the XPI of the antennas creates optimal conditions in the wireless MIMO channels. This combination does not only improve the system’s performance, but also turns the destructive physical phenomena in the spatial-domain, into an advantage. In this article, we focus on formulating a wireless communication MIMO model in millimeter-Wave (mmWave) channels under rain distortions. We demonstrate the optimal use of combining Orthogonal-Space-Time-Coding (OSTBC) and Maximal-Ratio-Receive-Combiner (MRRC) with cross-polarization diversity techniques, that utilize the tuning of the XPI. An analytical exact optimal solution is proposed, that allows the tuning of the leading parameters to achieve global optimal performance, in terms of channel-capacity and Bit-Error-Rate (BER). In addition, we propose a process of approximation of feedback-closed-loop-MIMO. The feedback is employed between the transmitter and the receiver, in the scenario of changes in the channel-response-matrix in-between successive symbol-times. The feedback was designed to acheive global-maximum channel-capacity, while preserving the channel-path orthogonality in order to minimize the BER.

## 1. Introduction

A wide variety of advanced wireless communication systems of outdoor-to-outdoor applications are being implemented and activated in the civilian and military fields. Examples of these systems are 5G-NR Heterogeneous Network (HetNet) [1,2], IAB in 5G-NR- mmWave networks Stand-Alone (SA) [3], None-Stand-Alone (NSA), small-cells 5G-NR and 6G wireless communication mobile relays [1,4]. The common goal of all these wireless systems is to achieve a comprehensive area coverage with significant QoS and ultra-high capacity for all OTA outdoor-to-outdoor users. Any comprehensive solution would have to deal with the following problems:deciding the relevant resolution (e.g., far field or close field, quantum or classical) in which the physical phenomena such as rain-distortion and urban-scattering-objects are described;formulating a communication model that takes into account the physical phenomena and the system mechanisms (e.g., antenna parameters, channel estimation, decoding algorithms and interference-cancellation algorithms);finding optimal solutions regarding ultra-capacity and minimal BER performance.

When transmitting considerable amounts of information on wireless channels under random physical destruction such as rich-scattering environments [5,6], multi-interference [7], path-loss Rayleigh–Fading environments, atmospheric and weather phenomena–the proposed method is desirable. Examples of these effects are rain distortion and scattering in urban environments [8] that cause severe attenuation of the transmitted signals in addition to significant fading effects at the receiver. Moreover, these effects cause transformation of the polarization components of the propagating electromagnetic wave in the media [9,10,11,12], including the outcome of None-Line-of-Site (NLOS). Considering rain distortions or urban objects means taking into account that a relationship exists between the input and the output linear depolarizing electric field components, propagating from the raindrop or the scattering objects in the medium. To illustrate the destructive rain phenomenon, it is possible to see the Figure 2e at [13], and the diagram at [9]. The effects of these phenomena on wireless communication systems based on orthogonal Dual-Polarization (DP) components, cause a deterioration in system performance, due to the sensitivity of orthogonality between polarized-signals and spatial-paths.

The most important phenomenon is scattering due to impact of the electromagnetic signal with random scattering objects or raindrops, under rich scattering environments, or under heavy rain, hail and snow events. To our best knowledge, the vast literature, in the case of heavy rain, hail, and snow, does not suggest any MIMO techniques that deal with these challenging scenarios. When the MIMO system contains no decorrelation mechanisms between sub-channels, significant fading and multipath effects are presented at the receiver side. In the case that the MIMO system contains decorrelation mechanisms, there is significant reduction in the diversity gain. What characterizes existing models of mmWave wireless communication systems are the classical LOS models. Thus, under the above-mentioned events, the LOS components are decomposed into NLOS, i.i.d, and Rayleigh MIMO channel components, with random multi-path effects, which are destructive to mmWave systems.

Another complicated problem is the loss of systemic energy efficiency. This power-loss is manifested in the fact that without any spatial techniques such as polarization diversity, direct receiving of a transmitted signal that transmits from a vertically polarized antenna and is received with a horizontally polarized antenna, and also in the opposite case (i.e., transmitting from horizontally polarized antenna and receiving with a vertically polarized antenna), that is, a cross-polarization path, will result in a significant loss of power of 3 dB or more in the received signal [14,15].

The second complicated problem regarding the issue of destruction effects, is the formulation of the communication model. The complexity of formulating a communication model involves identifying the statistical process of the channel. These processes are dependent and change rapidly according to rain intensity, scattering properties, density, rainfall rate, and drop radius and must be taken into account in the channel model. These statistical process also depends on the ratio between the drop size of the rain to the wavelength of the propagation wave and also depends on extreme rain rate conditions, including the rain velocity [16]. The model distribution converts from the known Rayleigh-Ricean distribution model to gamma distribution, or exponential distribution, or log-normal distribution [16].

The complex formulations of the proposed solutions are also a serious and complicated problem. It is because these effects, especially the non-locality of the rain distortion effects that are not local-spatial effects, can be bypassed, filtered spatially, or offset digitally, such as analog-digital-interference-cancellation [17,18,19], as can be applied to space interferences. These effects are randomly deployed across the entire area between the transmitter and the receiver, making it challenging to develop practical solutions to improve system performance.

Several recently developed techniques are accepted, implemented, and deal with destructive phenomena [10,20,21]. The common denominator of these techniques is the estimation of a uni-polarized correlated Rayleigh channel, with an approximation of a spatial correlation matrix. There is also sharing of overhead information between the receiver and the transmitter, such as correlation mechanisms (space and polarization), allowing for fixed receive or transmit power constraints, and allocating power to the uni-co and dual-polarized path schemes. In addition, it is possible to design intelligent [21,22], hybrid [21,23], and orthogonal-beam-forming [22] based on polarization schemes at the array antennas. However, the complexity and the rate of convergence of the approximation of the Channel-State-Information (CSI) relative to the real channel-response including the covariance-correlation polarization-matrix, as well as the architectural complexity required (RF-chains, phase-shiftier, amplifiers, and approximation of digital weight matrix) in both transmitter and receiver, may lead to a state of asymmetry of the cost versus effectiveness of these techniques, when dealing with destructive phenomena such as rain distortions.

In order to achieve these two performance objectives: ultra-capacity and the above-mentioned physical destructive scenarios, it is necessary to combine two advanced communication approaches. The first one is to achieve colossal bandwidth that is available only in the millimeter waveband that enables multi-gigabit wireless networks [8]. The second is to integrate practical advanced MIMO techniques such as Spatial-Multiplexing (SM) [24], Space-Time-Coding (STC) transmit-diversity [24,25,26], polarization-diversity [14,27,28], massive-MIMO [29] and hybrid-beam-forming [21,30].

In this article, we propose an innovative MIMO model for communication under destructive effects and we analyze the model for global-optimal-capacity in terms of XPI values. We also show that this process not only integrates well with various advanced MIMO existing techniques, but also reinforces them significantly. In addition to the analysis of the proposed communication model, we extract a range of XPI optimal parameters that lead to several optimal conditions of the system, such that the transmitter and receiver (that can be quite distant from each other, with different weather characteristics) can work with different XPI values and even achieve an optimal state. We have implemented wireless-communication outdoor-to-outdoor mm-Wave systems on the MATLAB SIMULINK platform. We show simulation results of the proposed global-optimization analysis combined with a few MIMO techniques.

In this article, we will also illustrate the damage from the rain phenomenon on the wireless channels in the perspective of sum-rate-capacity and BER. The essential affecting parameter on which our model is based, is defined in antenna theory as the XPI. This parameter’s natural and typical use is to isolate the cross-polar antenna pattern in a given polarization transmitter or receiver antenna. In wireless orthogonal polarization communication systems, the XPI cancels the leakage cross-polar effects (e.g., the power leakage from a vertically polarized transmitter antenna to a horizontally-polarized receiver antenna). This capability is critical in point-to-point back-haul mmWave-orthogonal polarization systems that create a few paths of Line-of-Site (LOS). However, in the real-destructive outdoor situations, such as rain effects, the rain creates cross-polar in the propagation channel paths and increases the cross-polar ratio (XPR) in the channel. The significance is that the LOS path splits, for example, from the vertical-transmission-antenna to the vertical-receiver-antenna, into two dependent polarized paths (the second path is from the vertical-transmission-antenna to the horizontal-receiver-antenna). This phenomenon also occurs in the dual-polarization case. In this state, we expand the use of XPI, increase the global-cross polarization in the channel, and achieve the capability of polarization-transmission diversity with the optimization and MIMO techniques.

The article is organized as follows: in Section 2, we present channel modeling of dual-polarization for MIMO systems and for Two-Input Two-Output (TITO) systems. In Section 3, the optimization of the channel MIMO-TITO capacity with dual-polarization and XPI power allocation under rain distortions, is presented. We describe implementation use of the optimal conditions through the reference model, that includes dual-polarization capability, but does not include any MIMO-diversity techniques. We compare the gain of the channel obtained from the efficient use of optimal parameters in the XPI mechanism and the growth of a channel-capacity from the use of the parameters at intermediate states and at destructive states in the XPI mechanism, causing poor system performance. The reference model forms a comparative basis instead of models that combine techniques with optimization in the XPI mechanism under variable wireless channel modes. These models are presented at the end of the section.

In Section 4, we optimize system performance by combining-OSTBC techniques and polarization-diversity with static–feedback in dual-polarization-MIMO channels with rain distortions. In Section 5, we describe the analysis of OFDM-dual-polarization and combined time-diversity under MIMO and mm-Wave channels with rain distortions.

Section 6 is devoted to the variation of channel matrix components in the transition between two successive symbol-times (from $t_{0}$ to $t_{0}+ts$) and the design of orthogonal conditions in the channel matrix with the help of a feedback. In this section, we present a significant BER and sum-rate-capacity performance integrating the use of the proposed feedback into the communication system and also comparing it with poor system performance caused by the non-integration of this feedback, all under the same challenging channel conditions. Finally, in Section 7, we derive some conclusions and a vision for further research.

## 2. Channel Modeling of Dual-Polarization for MIMO-TITO Systems

Many complex definitions, parameters, and characteristics of DP channels exist under different physical assumptions. A few basic definitions are needed to define a communication model that includes all these assumptions and complex parameters. For example, in practical antenna issues, the cross-polarization transmission-receiving (e.g., transmission signals from a vertically-polarized $T_{x}$ antenna, to a horizontally-polarized $R_{x}$ antenna) is a very significant definition. The direct meaning is perfect isolation between the orthogonal feeds at the antenna. The practical measurement of the extent of the depolarization in a wireless channel is defined as Cross-Polar Discrimination (XPD). Another example of a complex parameter is the XPI mechanism. The XPI describes the insufficient measure of energy given specific polarization and the relative part of the energy that decouples into the orthogonal polarization.

In the case of wireless-communication systems based on mmWave frequency bands and orthogonal polarization issues, the propagation waves that pass through the wireless media are very vulnerable. Under the assumption of flat-fading channels that includes destructive effects such as scatters, fading, interference, and additional weather effects, in the perspective of polarization related to the propagation channel, these effects are reflected and expressed as XPR. Both definitions, the XPI and the XPR, are combined and exist in the cross-polarization issue to global XPD. In general, high values of XPD would indicate a higher level of separability between the two states of polarization, and such channels are amenable to polarization multiplexing techniques. On the other hand, channels with lower values of XPD would indicate significant cross-coupling between the two states of polarization and encourage MIMO-diversity techniques.

In this section, we describe a general MIMO-TITO channel modeling under rain or scatter distortion. To simplify the channel model, we consider the 2×2 MIMO-TITO assumption, flat-fading channels, and outdoor-to-outdoor scenario. This general MIMO-TITO channel model expresses the media between transmitter-receiver wireless communication systems based on an array of orthogonal dual-polarization antennas. The array antennas at the transmitter includes one vertical-linear polarization and one horizontal-linear polarization. On the receiver side, we assume a symmetric array antenna as at the transmitter side with co-polarization at the transmitter-receiver antennas. This architecture creates a two LOS environment based on orthogonal polarization separability conditions at the wireless channel that ideally offers a much better separation between channels. Under ideal conditions on the channel, we assume Ricean distribution with NLOS components close to zero. However, as we mentioned, in the case of present destructive effects, the generally MIMO-TITO channel modeling included NLOS Rayleigh components.

The general communication MIMO-TITO model combining components of LOS and components of NLOS is described as:(1)$$\begin{array}{c} Y=\sqrt{P}HS+Z \end{array}$$
where *P* is the total transmission power, *S* is the symbol-matrix transmission, *Z* is the independent complex Gaussian random variable noise, and *H* is the MIMO channel response matrix under Quasistatic Flat–Ricean–Rayleigh–Fading assumptions, given by:(2)$$\begin{array}{c} H=\underset{LOS}{\underbrace{\sqrt{\frac{k}{1+k}}\cdot R\cdot \left[\begin{array}{cc} \exp\left(j\Psi \right) & 0\\ 0 & \exp\left(j\Psi \right) \end{array}\right]\cdot T}}+\underset{NLOS}{\underbrace{\sqrt{\frac{1}{1+k}}\cdot R\cdot H_{medium}\cdot T}} \end{array}$$
where Ψ is the orientation angle between the transmitter-receiver antennas, and *k* is the Ricean–Fading distribution factor characterized. $H_{medium}$ is the MIMO channel response that describes the NLOS components, *T* and *R*, are the coupling antenna matrices at the transmitter and the receiver sides, respectively, and will be defined going forward. Suppose that Ψ=0, the orientation matrix that describes the LOS components will be equal to the identity matrix, *I*. Our assumption in this article is that with the presence of the destructive effect, for example, rain distortion, the Ricean–Fading distribution factor, *k*, is equal to zero, and $H_{medium}$ becomes the MIMO channel response channel under Rayleigh–Fading assumptions, which is defined by:(3)$$\begin{array}{c} H_{medium}=\left[\begin{array}{cc} h_{xx} & h_{xy}\\ h_{yx} & h_{yy} \end{array}\right] \end{array}$$

The components, $h_{xy},h_{yx}$ describe the cross-dual-polarization paths between the vertically polarized transmission antenna to the horizontally polarized receiver antenna and the horizontally polarized transmission antenna to the vertically polarized receiver antenna, respectively. The components, $h_{xx},h_{yy}$ define the co-polarization paths between the vertically polarized transmission antenna to the vertically polarized receiver antenna and the horizontally polarized transmission antenna to the horizontally polarized receiver antenna, respectively. To describe the polarization phenomenon in these channels, we will define basic parameters and break down the general channel response matrix into several matricial components based on these parameters. A cross-polar antenna pattern with finite XPI can be defined by a coupling antenna matrix at the transmitter side and also at the receiver side as: $$\begin{array}{cc} & T=\left[\begin{array}{cc} \sqrt{1-\frac{1}{XPI_{T}}} & \sqrt{\frac{1}{XPI_{T}}}\\ \sqrt{\frac{1}{XPI_{T}}} & \sqrt{1-\frac{1}{XPI_{T}}} \end{array}\right]\\ and,\\ & R=\left[\begin{array}{cc} \sqrt{1-\frac{1}{XPI_{R}}} & \sqrt{\frac{1}{XPI_{R}}}\\ \sqrt{\frac{1}{XPI_{R}}} & \sqrt{1-\frac{1}{XPI_{R}}} \end{array}\right] \end{array}$$
where the scalar XPI, can obtain the values between 1≤XPI≤∞. When XPI=∞, this value represents perfect polar power isolation. This mean that in an antenna defined with vertical polarization, all the signal’s power is projected through the vertical polarization direction without leakage towards the horizontal polarization. When the antenna is defined as horizontal polarization, all the signal’s power is projected through the horizontal polarization direction without leakage towards the vertical polarization. If we suppose that XPI is close to 1, the transmitted or received signal power is divided evenly between the vertical and horizontal polarization in the vertical antenna. The same division process also occurs in the horizontal antenna. This situation yields DP with a cross-polarization MIMO model. $XPI_{T}$ and $XPI_{R}$ are defined at the transmitter and receiver antennas, respectively. $\delta_{xy}$, and $\delta_{yx}$ is the complex factor cross-polarization that is created from the rain cases or other destructive effects and defined as: $$\begin{array}{cc} & \delta_{xy}=\frac{h_{xy}}{h_{xx}}\\ and,\\ & \delta_{yx}=\frac{h_{yx}}{h_{yy}} \end{array}$$

The XPR is a real value that describes the ratio of the channels norms (i.e., the gain on the channel i,j represented by ${\left|h_{i=x,j=y}\right|}^{2}$), denoted as: $$\begin{array}{cc} & XPR_{x}=\frac{1}{{\left|\delta_{xy}\right|}^{2}}\\ and,\\ & XPR_{y}=\frac{1}{{\left|\delta_{yx}\right|}^{2}} \end{array}$$

The new $H_{medium}$, in relation to the $H_{medium}$ in (Equation 2), is:(4)$$\begin{array}{c} H_{medium}=\left[\begin{array}{cc} h_{xx} & \delta_{x,y}h_{xx}\\ \delta_{y,x}h_{yy} & h_{yy} \end{array}\right] \end{array}$$

Rewriting
(5)$$\left\{\begin{array}{c} \frac{1}{XPI{}_{T}}=\alpha ,\frac{1}{XPI{}_{R}}=\beta \\ \sqrt{\alpha }=\sin\left(\theta \right),\sqrt{\beta }=\sin\left(\phi \right)\\ \sqrt{1-\alpha }=\cos\left(\theta \right),\sqrt{1-\beta }=\cos\left(\phi \right) \end{array}\right.$$
where
$$\begin{array}{cc} H{}_{TR} & =\left(\sqrt{\frac{1}{1+k}}\right)RH_{medium}T=\\ & =\left(\sqrt{\frac{1}{1+k}}\right)\left[\begin{array}{cc} \cos\left(\theta \right) & \sin\left(\theta \right)\\ \sin\left(\theta \right) & \cos\left(\theta \right) \end{array}\right]\left[\begin{array}{cc} h_{xx} & \delta_{x,y}h_{xx}\\ \delta_{y,x}h_{yy} & h_{yy} \end{array}\right]\left[\begin{array}{cc} \cos\left(\phi \right) & \sin\left(\phi \right)\\ \sin\left(\phi \right) & \cos\left(\phi \right) \end{array}\right] \end{array}$$

The channel response matrix, $H_{{}_{TR}}$, now represents the DP channel-matrix combining spatial-separation, dual-polarized arrays, global XPD, and finite XPI under Quasistatic Flat–Rayleigh–Fading assumption.

Defining
(6)$$\left\{\begin{array}{c} A=\cos\left(\theta \right)\sin\left(\phi \right)\\ B=\cos\left(\theta \right)\cos\left(\phi \right)\\ C=\sin\left(\theta \right)\cos\left(\phi \right)\\ D=\sin\left(\theta \right)\sin\left(\phi \right), \end{array}\right.$$

$H_{{}_{TR}}$ now define as: $$\begin{array}{cc} & H_{{}_{TR}}=\left(\sqrt{\frac{1}{1+k}}\right)\left[\begin{array}{cc} h_{xx}\left(B+\delta_{x,y}A\right)+h_{yy}\left(D+\delta_{y,x}C\right) & h_{xx}\left(A+\delta_{x,y}B\right)+h_{yy}\left(C+\delta_{y,x}D\right)\\ h_{xx}\left(C+\delta_{x,y}D\right)+h_{yy}\left(A+\delta_{y,x}B\right) & h_{xx}\left(D+\delta_{x,y}C\right)+h_{yy}\left(B+\delta_{y,x}A\right) \end{array}\right] \end{array}$$

As we mentioned and will demonstrate in the following sections, the polarization state in the channel can be influenced by tuning the XPI mechanism in the receiver and transmission antennas. This process is significant and necessary because the interference’s effects, and physical phenomena, such as rain, constantly change the channel’s conditions and affect the system’s performance. In this situation, optimizing the system performance by tuning the XPI and combining advanced MIMO techniques is critical.

## 3. Optimization of Channel MIMO Capacity with Dual-Polarization and XPI Power Allocation

In this section, we analyze and simulate the effect of rain distortion in 2×2 MIMO-TITO DP wireless communication systems, without any diversity techniques, on the channel capacity. We consider these processes as a reference model to lay the first milestone for presenting a base of conflict from the great ones in DP MIMO-wireless communication theory.

The following facts make up the basis of the conflict: The first one, when the modern communication MIMO-system is combined with polarization diversity techniques, the MIMO channels with polarization diversity achieve a very low correlation between the sub-channels, which is beneficial for its capacity. Simultaneously, the spatial multi-paths created from the spatial-diversity techniques are extraordinarily vulnerable and experience decline in the Signal-to-Noise-Ratio (SNR), for example, in the diversity losses. This situation not only leads to detrimental BER performance but also affects the channel capacity and can lead to deterioration in the sum-rate capacity. However, the second fact is based on the scenario (proactively or in a physical design) that leads to a situation with high XPD. The high value of XPR helps to achieve better capacity for dual-polarized MIMO channels and create a few very good Single-Input-Single-Output (SISO) links.

The conflict (in the perspective of DP) is considered when there is a destructive effect such as rain distortion. On the one hand, if we implement polarization diversity or transmit diversity in wireless MIMO-communication systems, the sub-channel power losses, in addition to the diversity losses, are, in general, detrimental to the capacity of MIMO channels (Although, in this situation, there is an increase in the rank order of the channel matrix). However, on the other hand, these configurations achieve a very low correlation between the sub-channels, which is beneficial to achieving common BER values. On the other side of the conflict, if we concentrate energy in the apparent direction of co-polarization without any diversity techniques, in addition to the increase in the bit-rate transmission compared to the OSTBC, for example (a situation that leads to an increase in the sum-rate capacity and the XPR value), the strong correlations can lead to a significant decrease in the BER performance (In addition to the lack of capabilities to deal with multi-paths phenomena).

In this article, we not only settle for presenting the conflict, but we also offer a few solutions for resolving the conflict. Without any diversity techniques, this reference model, described in this section, will form a comparative basis against the model that includes the diversity method described in Section 4.

The first step in the process of optimizing the MIMO channel capacity, is to extract from (2) the optimal values in terms of XPI values, where the function to be maximized is the square of the Frobenius norm of the $H_{TR}$ that leads to optimal capacity. In Section 4.2, we find analytically 4 points of global maximum of the above-mentioned function. As the function is shown to be convex, we also find its global maximum (related to the worst destructed channel). Note that the mathematical analysis described in Section 4.2, is more general than needed here (since the $H_{TR}$ matrix there is 4×2 resulted from the application of the OSTBC), but its results are obviously applicable also here.

By tuning the XPI mechanism in the transmitter and the receiver to the optimal values, it is possible to design the channels as optimal MIMO-TITO channels, as is seen in the following Table 1. The table contains all the optimal channels with their related XPI values. The same optimal values are formed using matrices *T* and *R*:

In the second step, we propose a design of a simulator scheme based on the SIMULINK/MATLAB platform. This simulator includes a transmitter-receiver OFDM mmWave architecture under wireless MIMO-TITO channels with rain distortion parameters based on [16]. The description of the general blocks scheme included the network wireless configuration is shown in Figure 1, and characterizing parameters are described in Table 1.

The blocks scheme describe in Figure 1 and Figure 2, included at the transmitter side, with the data-source block streaming data to the QPSK modulator. From the QPSK, the base-band data is split into two different OFDM transmitter schemes. These two transmitter paths stream the base-band data to two independent RF chain blocks. The blocks include the mmWave up-convert block, XPI mechanism, and two orthogonal (vertical-horizontal-linear-polarization) polarization transmission antennas. The block transmission, radiator, and propagation pass the 2×2 MIMO-TITO-channel matrix under the assumption of Quasistatic Flat-Rayleigh–Fading and rain distortion. On the receiver side, a collector RF block is situated at the receiver front. This block included two orthogonal (vertical-horizontal-linear-polarization) polarization antenna receivers, an XPI mechanism, and two RF chains, including a down-convert block. The data decoding passes through the OFDM receiver block and is decoded by QPSK demodulator. (Please note that the units and parameters used in the conducted simulations are described in Table 2).

The MIMO-channel-capacity under the assumption of channel-state-information-at receiver only (CSIR) follows as [24,31]:(7)$$\begin{array}{cc} C_{csi} & =E\left[log_{2}det\left(I_{N_{Y}}+\frac{P_{t}}{N_{t}\sigma_{n}^{2}}HH^{*}\right)\right]=E\left[\sum_{i=1}^{N_{Y}}log_{2}\left(1+\frac{P_{t}}{N_{t}\sigma_{n}^{2}}\lambda_{i}^{2}\right)\right] \end{array}$$
where $N_{Y}$ is the number of the receive antennas, $P_{t}$ is the total transmit power, $\sigma_{n}^{2}$ is the variance of the independent complex Gaussian random variable noise, and $\lambda_{i}^{2}$ is the i’th eigenvalue of $HH^{*}$.

In Figure 3, numerical-results of the proposed simulator, i.e., simulation of MIMO-channel-capacity vs. $\frac{E_{b}}{N_{0}}$ (Ratio of energy per bit to white Gaussian noise density), in addition to rain distortion effects are illustrated. The red, green, and blue graphs describe the four points of global maximum, intermediate value, and most distortion value, respectively. Note that the simulation result of all four points of global maximum is identical. From the analysis of the results, one can deduce the first two main insights: The first one, the simulation results, represents the analytical analysis we performed. The red graph describes the maximum average capacity of the wireless communication system. By optimizing XPI values under rain distortion, we design the optimal conditions to obtain the total capacity. The green graph represents the intermediate values of the optimization results ($\phi =\frac{\pi }{8},\theta =\frac{3\pi }{8}$). Note that in this situation, the performance of the capacity channel is still high relative to the blue graph. Similar to the red graph, and in contrast to the blue graph, the green graph increases with an increase in the value of $\frac{E_{b}}{N_{0}}$. The blue graph represents the maximum shortfall capacity performance led by the most destructive value ($\phi =\frac{\pi }{4},\theta =\frac{\pi }{4}$). The second insight is the relatively constant behavior of the blue graph. The constant behavior of the blue graph is reflected starting from the value of $\frac{E_{b}}{N_{0}}∼14$ dB, while in these values in the green and red graphs, there is an increase to significantly higher values. The direct conclusion is that in the presence of a destructive effect on the wireless channel, improper tuning of an XPI mechanism leads to lousy system performance.

## 4. Optimization of Channel MIMO Capacity by Combining OSTBC and Polarization-Diversity with XPI

This section describes the effects and benefits of adding transmission-time-polarization-diversity MIMO techniques to the communication model under rain distortion. This combination is possible with the coding of the OSTBC on the transmitter side, the MRRC technique on the receiver side, and an XPI power allocation mechanism in the transmitter-receiver antenna array.

Our approach is to create a comparative basis of the system’s performance between the relative communication model discussed in Section 3 and the model discussed in this section. From this basis, it will be possible to draw significant insights to deal with these challenging scenarios.

We assume that the same optimization processes described in Section 3 are also implemented in this analysis. The optimized channel values are based on Table 1, and the architecture of the transmitter-receiver is described in Figure 4. Note that the OSTBC block is implemented between the QPSK modulator and the OFDM scheme transmitter (The OSTBC operation will be explained in the following section).

This section is divided into Section 4.1, Section 4.2 and Section 4.3: the Section 4.1 describes the OSTBC operation and the MRRC-combiner. The Section 4.2 describes the performance of OSTBC–OFDM combined with the DP model under rain distribution. The Section 4.3 describes numerical results of MIMO-channel-capacity vs. $\frac{E_{b}}{N_{0}}$, under combining OSTBC operation to the transmitter, optimizing XPI value, and MRRC decoding in the receiver, under the same rain conditions as described in Section 3.

### 4.1. Diversity Transmission Using OSTBC-MRRC MIMO Technique

To understand the OSTBC operation in depth, we will briefly explain the known 2×2 OSTBC technique [25]. The transmission consists of two transmission antennas, $TX_{1}$ and $TX_{2}$. Next, the signal transmits through a MIMO channel model with four flat fading paths independent and uncorrelated. These four paths are represented by complex channel gains $h_{11}$, $h_{12}$, $h_{21}$, $h_{22}$. The indexes are the indexes of TX and RX antennas, respectively, as shown in Table 3 [24,25]. The receive block includes two receive antennas, $RX_{1}$ and $RX_{2}$, and an OSTBC decoder combined with MRRC or diversity combining channel state estimation with Maximum-Likelihood (ML) decoding.

The OSTBC operation of the two symbols $s_{1}$ and $s_{2}$ is defined in Table 4 and includes two transmit antennas under two-time slots. At some time *t*, symbols $s_{1}$ and $s_{2}$ are transmitted from $TX_{1}$ and $TX_{2}$, respectively. At time $t+T_{s}$, where $T_{s}$ is the symbol duration, the symbols $-s_{2}^{*}$ and $s_{1}^{*}$ are transmitted from $TX_{1}$ and $TX_{2}$, respectively [25]. The operator ${\left(\;\cdot \;\right)}^{*}$ represents the complex conjugate.

The received signal *Y* and Additive White Gaussian Noise (AWGN) signal *Z* are denoted by two indexes, $Y_{k}\left(l\right)$ and $Z_{k}\left(l\right)$, where k=1,2 denotes the number of received antennas, and l=1,2 denotes the received signal at time *t* or $t+T_{s}$, respectively. In Table 5, for example, $Y_{1}\left(1\right)$ denotes the received signal at time *t* at $RX_{1}$.

The receiver equations are [24,25]
(8)$$\begin{array}{cc} \; & Y_{1}\left(1\right)=h_{11}s_{1}+h_{12}s_{2}+Z_{1}\left(1\right)\\ \; & Y_{1}\left(2\right)=-h_{11}s_{2}^{*}+h_{12}s_{1}^{*}+Z_{1}\left(2\right)\\ \; & Y_{2}\left(1\right)=h_{21}s_{1}+h_{22}s_{2}+Z_{2}\left(1\right)\\ \; & Y_{2}\left(2\right)=-h_{21}s_{2}^{*}+h_{22}s_{1}^{*}+Z_{2}\left(2\right) \end{array}$$
The combining rules for a 2×2 OSTBC system are [24,25]
(9)$$\begin{array}{cc} \; & {\tilde{s}}_{1}=h_{11}^{*}Y_{1}\left(1\right)+h_{12}Y_{1}^{*}\left(2\right)+h_{21}^{*}Y_{2}\left(1\right)+h_{22}Y_{2}^{*}\left(2\right)\\ \; & {\tilde{s}}_{2}=h_{12}^{*}Y_{1}\left(1\right)-h_{11}Y_{1}^{*}\left(2\right)+h_{22}^{*}Y_{2}\left(1\right)-h_{21}Y_{2}^{*}\left(2\right) \end{array}$$
Substituting (Equation 8) into (Equation 9) yields [24]
$$\begin{array}{ccc} {\tilde{s}}_{1} & = & \left(\sum_{i=1}^{2}\sum_{j=1}^{2}{\left|h_{ij}\right|}^{2}\right)s_{1}+h_{11}^{*}z_{1}\left(1\right)+h_{12}z_{1}^{*}\left(2\right)+h_{21}^{*}z_{2}\left(1\right)+h_{22}z_{2}^{*}\left(2\right) \end{array}$$
and
(10)$$\begin{array}{ccc} {\tilde{s}}_{2} & = & \left(\sum_{i=1}^{2}\sum_{j=1}^{2}{\left|h_{ij}\right|}^{2}\right)s_{2}+h_{12}^{*}z_{1}\left(1\right)-h_{11}z_{1}^{*}\left(2\right)+h_{22}^{*}z_{2}\left(1\right)-h_{21}z_{2}^{*}\left(2\right) \end{array}$$
The estimation of each symbol $s_{1}$ and $s_{2}$ is multiplied by the norm of each complex channel gain without the other symbol’s presence. Finally, the last receive step is the ML estimate. The transmitted symbol is estimated as follows [24]:(11)$$\begin{array}{cc} \; & \hat{s_{1}}=argmin_{s_{1}}\left[{\left|\tilde{s_{1}}-s_{1}\right|}^{2}\right]\\ \; & \hat{s_{2}}=argmin_{s_{2}}\left[{\left|\tilde{s_{2}}-s_{2}\right|}^{2}\right] \end{array}$$
Another means to display Equation (Equation 8) is through matrices. After mathematical operations of conjugation, the model will be [24,25]
(12)$$\begin{array}{c} Y=\left[\begin{array}{c} Y_{1}\left(1\right)\\ Y_{1}^{*}\left(2\right)\\ Y_{2}\left(1\right)\\ Y_{2}^{*}\left(2\right) \end{array}\right]=\left[\begin{array}{c} h_{11}\;\;h_{12}\\ h_{12}^{*}\;-h_{11}^{*}\\ h_{21}\;\;h_{22}\\ h_{22}^{*}\;-h_{21}^{*} \end{array}\right]\left[\begin{array}{c} s_{1}\\ s_{2} \end{array}\right]+\left[\begin{array}{c} z_{1}\left(1\right)\\ z_{1}^{*}\left(2\right)\\ z_{2}\left(1\right)\\ z_{2}^{*}\left(2\right) \end{array}\right] \end{array}$$
In short,
(13)$$\begin{array}{c} Y=H_{TR}S+Z \end{array}$$
where *S* is the OSTBC matrix symbol, *Z* is the independent complex Gaussian random variable noise, and $H_{TR}$ is the MIMO channel response matrix under Quasistatic Flat–Rayleigh–Fading assumptions. In addition, $H_{TR}$ is:(14)$$\begin{array}{c} H_{TR}=\left[\begin{array}{c} h_{11}\;\;h_{12}\\ h_{12}^{*}\;-h_{11}^{*}\\ h_{21}\;\;h_{22}\\ h_{22}^{*}\;-h_{21}^{*} \end{array}\right] \end{array}$$
An important feature of this OSTBC coding matrix is the orthogonality between its columns [24]. If the condition
(15)$$column_{1}^{H}column_{2}=0$$
is met, the column vectors are orthogonal. The operator ${\left(\;\cdot \;\right)}^{H}$ is the complex transpose. For the $H_{TR}$ case,
$$\begin{array}{c} \left[\begin{array}{cccc} h_{11}^{*} & \;h_{12} & \;h_{21}^{*} & \;h_{22} \end{array}\right]\cdot \left[\begin{array}{c} \;h_{12}\\ -h_{11}^{*}\\ \;h_{22}\\ -h_{21}^{*} \end{array}\right]=0 \end{array}$$
This critical feature allows the symbols to be decoded and Equation (Equation 10) to be obtained. It can also be seen by presenting Equation (Equation 9) as the following matrix:$$\begin{array}{c} \tilde{S}={\left(H_{TR}\right)}^{H}Y \end{array}$$
Next, (Equation 13) is substituted into the above equation to obtain
$$\begin{array}{c} \tilde{S}={\left(H_{TR}\right)}^{H}\left(H_{TR}S+Z\right)=\sum_{i=1}^{2}\sum_{j=1}^{2}{\left|h_{ij}\right|}^{2}I_{2}S+{\left(H_{TR}\right)}^{H}Z \end{array}$$
which is the same as Equation (Equation 10). $I_{2}$ denotes an identity matrix of size 2.

### 4.2. Performance of OSTBC–OFDM Combined with Dual-Polarization Model, under Rain Distortion in mmWave Channels

In this section, we formulate and demonstrate the formulation and optimization of the communication model described in Section 2, in combination with OSTBC-OFDM, XPI, and polarization techniques.

The OSTBC operation on $H_{TR}$ leads to:$$\begin{array}{c} H_{TR}=\left[\begin{array}{cc} \left(h_{xx}\left(B+\delta_{x,y}A\right)+h_{yy}\left(D+\delta_{y,x}C\right)\right) & \left(h_{xx}\left(A+\delta_{x,y}B\right)+h_{yy}\left(C+\delta_{y,x}D\right)\right)\\ {\left(h_{xx}\left(A+\delta_{x,y}B\right)+h_{yy}\left(C+\delta_{y,x}D\right)\right)}^{*} & -{\left(h_{xx}\left(B+\delta_{x,y}A\right)+h_{yy}\left(D+\delta_{y,x}C\right)\right)}^{*}\\ \left(h_{xx}\left(C+\delta_{x,y}D\right)+h_{yy}\left(A+\delta_{y,x}B\right)\right) & \left(h_{xx}\left(D+\delta_{x,y}C\right)+h_{yy}\left(B+\delta_{y,x}A\right)\right)\\ {\left(\left(h_{xx}\left(D+\delta_{x,y}C\right)+h_{yy}\left(B+\delta_{y,x}A\right)\right)\right)}^{*} & -{\left(\left(h_{xx}\left(C+\delta_{x,y}D\right)+h_{yy}\left(A+\delta_{y,x}B\right)\right)\right)}^{*} \end{array}\right] \end{array}$$

If we neglect the power leakage $XPI_{T},XPI_{R}$, meaning A=C=D=0, and B=1, the channel response matrix, $H_{{}_{TR}}$ becomes (Equation 4).

To optimize the system performances, for example, the BER, energy efficiency, spatial–separation, and also to maximize the capacity of the throughput with DP, and to add precision to the process of the CSIR estimation, the receiver must approximate three different stages one by one. The first stage is to estimate the MIMO-multi-path fading response channel, ${\tilde{H}}_{TR}$. The next step is to generate a receiver-transmitter convention on one of the four optimal cases (according to the optimization process we will present below) and to adjust the XPI mechanism in the receiver-transmitter to meet the selected condition. Finally, it is necessary to measure parameters that we are interested in improving in the receiver.

Let
(16)$$\begin{array}{cc} \; & f\left(\theta ,\phi \right)=trace\left(H_{TR}^{*}H_{TR}\right)=\\ \; & =2\left(h_{x,x}\left(B\delta_{x,y}+A\right)+h_{y,y}\left(D\delta_{y,x}+C\right)\right)\cdot \left(h_{x,x}^{*}\left(B\delta_{x,y}^{*}+A\right)+h_{y,y}^{*}\left(D\delta_{y,x}^{*}+C\right)\right)+\\ \; & +2\left(h_{x,x}\left(A\delta_{x,y}+B\right)+h_{y,y}\left(C\delta_{y,x}+D\right)\right)\cdot \left(h_{x,x}^{*}\left(A\delta_{x,y}^{*}+B\right)+h_{y,y}^{*}\left(C\delta_{y,x}^{*}+D\right)\right)+\\ \; & +2\left(h_{y,y}\left(B\delta_{y,x}+A\right)+h_{x,x}\left(D\delta_{x,y}+C\right)\right)\cdot \left(h_{y,y}^{*}\left(B\delta_{y,x}^{*}+A\right)+h_{x,x}^{*}\left(D\delta_{x,y}^{*}+C\right)\right)+\\ \; & +2\left(h_{y,y}\left(A\delta_{y,x}+B\right)+h_{x,x}\left(C\delta_{x,y}+D\right)\right)\cdot \left(h_{y,y}^{*}\left(A\delta_{y,x}^{*}+B\right)+h_{x,x}^{*}\left(C\delta_{x,y}^{*}+D\right)\right)=\\ \; & =2{\left|h_{x,x}\left(B\delta_{x,y}+A\right)+h_{y,y}\left(D\delta_{y,x}+C\right)\right|}^{2}+\\ \; & +2{\left|h_{x,x}\left(A\delta_{x,y}+B\right)+h_{y,y}\left(C\delta_{y,x}+D\right)\right|}^{2}+\\ \; & +2{\left|h_{y,y}\left(B\delta_{y,x}+A\right)+h_{x,x}\left(D\delta_{x,y}+C\right)\right|}^{2}+\\ \; & +2{\left|h_{y,y}\left(A\delta_{y,x}+B\right)+h_{x,x}\left(C\delta_{x,y}+D\right)\right|}^{2}=\\ \; & =2{∥\left[\begin{array}{c} h_{x,x}\left(B\delta_{x,y}+A\right)+h_{y,y}\left(D\delta_{y,x}+C\right)\\ h_{x,x}\left(A\delta_{x,y}+B\right)+h_{y,y}\left(C\delta_{y,x}+D\right)\\ h_{y,y}\left(B\delta_{y,x}+A\right)+h_{x,x}\left(D\delta_{x,y}+C\right)\\ h_{y,y}\left(A\delta_{y,x}+B\right)+h_{x,x}\left(C\delta_{x,y}+D\right) \end{array}\right]∥}^{2}. \end{array}$$

Then, in order to maximize the capacity (as well as minimizing the BER), we need to maximize *f*. Now, we will show that *f* is a convex function, as a function of A,B,C,D over the region 0≤A,B,C,D≤1.

**Lemma** **1.**
*Let $g:C^{m}\to R$ be defined by $g\left(v\right)={∥v∥}^{2}$. Then, g is convex.*


**Proof.** Let $v_{1},v_{2}\in C^{m}$ and let 0≤λ≤1. We need to show that $g\left(\lambda v_{1}+\left(1-\lambda \right)v_{2}\right)\leq \lambda g\left(v_{1}\right)+\left(1-\lambda \right)g\left(v_{2}\right)$. Now,
$$\begin{array}{cc} & g\left(\lambda v_{1}+\left(1-\lambda \right)v_{2}\right)={∥\lambda v_{1}+\left(1-\lambda \right)v_{2}∥}^{2}\\ & =\lambda^{2}{∥v_{1}∥}^{2}+2\lambda \left(1-\lambda \right)ℜ\left(\langle v_{1},v_{2}\rangle \right)+{\left(1-\lambda \right)}^{2}{∥v_{2}∥}^{2}\\ & \leq \lambda^{2}{∥v_{1}∥}^{2}+2\lambda \left(1-\lambda \right)\left|\langle v_{1},v_{2}\rangle \right|+{\left(1-\lambda \right)}^{2}{∥v_{2}∥}^{2}\\ & \leq \lambda^{2}{∥v_{1}∥}^{2}+2\lambda \left(1-\lambda \right)∥v_{1}∥∥v_{2}∥+{\left(1-\lambda \right)}^{2}{∥v_{2}∥}^{2}, \end{array}$$
where in the last passage, we used Cauchy-Swartz inequality. Since
$$\lambda g\left(v_{1}\right)+\left(1-\lambda \right)g\left(v_{2}\right)=\lambda {∥v_{1}∥}^{2}+\left(1-\lambda \right){∥v_{2}∥}^{2},$$
it is therefore sufficient to prove that:
$$\lambda^{2}{∥v_{1}∥}^{2}+2\lambda \left(1-\lambda \right)∥v_{1}∥∥v_{2}∥+{\left(1-\lambda \right)}^{2}{∥v_{2}∥}^{2}\leq \lambda {∥v_{1}∥}^{2}+\left(1-\lambda \right){∥v_{2}∥}^{2}.$$
Indeed,
$$\begin{array}{cc} & \lambda^{2}{∥v_{1}∥}^{2}+2\lambda \left(1-\lambda \right)∥v_{1}∥∥v_{2}∥+{\left(1-\lambda \right)}^{2}{∥v_{2}∥}^{2}\leq \lambda {∥v_{1}∥}^{2}+\left(1-\lambda \right){∥v_{2}∥}^{2}\\ & \Leftrightarrow 0\leq \lambda \left(1-\lambda \right){∥v_{1}∥}^{2}-2\lambda \left(1-\lambda \right)∥v_{1}∥∥v_{2}∥+\lambda \left(1-\lambda \right){∥v_{1}∥}^{2}\\ & \Leftrightarrow 0\leq {∥v_{1}∥}^{2}-2∥v_{1}∥∥v_{2}∥+{∥v_{1}∥}^{2}\\ & \Leftrightarrow 0\leq {\left(∥v_{1}∥-∥v_{2}∥\right)}^{2}. \end{array}$$This completes the proof. □

**Lemma** **2.**
*Let $h:R^{n}\to C^{m}$ denote any linear transformation and let $g:C^{m}\to R$ be defined by $g\left(v\right)={∥v∥}^{2}$. Let $f:R^{n}\to R$ be defined by $f\left(x\right)=g\left(h\left(x\right)\right)$. Then, f is convex.*


**Proof.** Let $x_{1},x_{2}\in R^{n}$ and let 0≤λ≤1. Let $v_{1}=h\left(x_{1}\right)$ and $v_{2}=h\left(x_{2}\right)$. Then,
$$\begin{array}{cc} & f\left(\lambda x_{1}+\left(1-\lambda \right)x_{2}\right)=g\left(h\left(\lambda x_{1}+\left(1-\lambda \right)x_{2}\right)\right)\\ & =g\left(\lambda h\left(x_{1}\right)+\left(1-\lambda \right)h\left(x_{2}\right)\right)\\ & =g\left(\lambda v_{1}+\left(1-\lambda \right)v_{2}\right)\\ & \leq \lambda g\left(v_{1}\right)+\left(1-\lambda \right)g\left(v_{2}\right)\\ & =\lambda g\left(h\left(x_{1}\right)\right)+\left(1-\lambda \right)g\left(h\left(x_{2}\right)\right)\\ & =\lambda f\left(x_{1}\right)+\left(1-\lambda \right)f\left(x_{2}\right), \end{array}$$
where we used the linearity of *h* and Lemma 1. □

We therefore conclude that the function *f* given by (Equation 16) is a convex function as a function of A,B,C,D over the region 0≤A,B,C,D≤1. By a well known theorem, the maximum of *f* is achieved on the exposed points of the region; that is, on one of the 16 nodes $A,B,C,D\in \left\{0,1\right\}$. However, since here A,B,C,D depend on θ,ϕ, we have only 4 cases to consider:(17)$$\begin{array}{cc} \; & \mathrm{case}1:\theta =0,\phi =0\Rightarrow A=0,B=1,C=0,D=0\\ \; & \mathrm{case}2:\theta =0,\phi =\frac{\pi }{2}\Rightarrow A=1,B=0,C=0,D=0\\ \; & \mathrm{case}3:\theta =\frac{\pi }{2},\phi =0\Rightarrow A=0,B=0,C=1,D=0\\ \; & \mathrm{case}4:\theta =\frac{\pi }{2},\phi =\frac{\pi }{2}\Rightarrow A=0,B=0,C=0,D=1. \end{array}$$

### 4.3. Numerical Results of MIMO-Channel-Capacity, with OSTBC and MRRC, under Rain Distortions, Optimized by XPI Values

In Figure 5, numerical results of the MIMO channel capacity of a communication system in mmWave with a combination of diversity techniques, XPI mechanism, under effects of rain distortion versus $\frac{E_{b}}{N_{0}}$ are illustrated. This section refers to MIMO capabilities regarding OSTBC-MRRC diversification techniques according to the optimal XPI values given in (Equation 17). These optimal cases create polarization diversity and integrate with the application of OSTBC-MRRC techniques. Diversity techniques result in a low correlation between the DP MIMO paths. Thus, it is reasonable to assume that all the channel paths are independent of each other and can reduce the BER relative to system performance without the combination of these techniques (as we will see in Section 5). On the other hand, as we have seen in Section 4.2, the transmission of the same two symbols in two cycles of time causes a decrease in the transmission bit rate. This situation results in a reduction of the average channel capacity relative to the case without any diversity.

Note that if for example we transmit power in directions u,v such that u⊥v, with powers $P_{u},P_{v}$ resp. then, the total received power is $\sqrt{P_{u}^{2}+P_{v}^{2}}$ that becomes $P_{u}$ if the direction *v* undergoes total destruction. However, if the directions are not perpendicular and have an angle 0<γ<π/2 between them then, the total received power is $\sqrt{{\left(P_{u}+P_{v}\cos\left(\gamma \right)\right)}^{2}+{\left(P_{v}\sin\left(\gamma \right)\right)}^{2}}$, and under the above assumption, the destruction now is in direction $v-\langle v,u\rangle u$ that is perpendicular to *u*. Therefore, the value $P_{v}\sin\left(\gamma \right)$ is lost and the total received power now is $P_{u}+P_{v}\cos\left(\gamma \right)>P_{u}$. Therefore, the capacity in (Equation 11) lowers-down in the case of perpendicular paths, when part of the paths undergoes total destruction, as is seen from the comparison between Figure 3 and Figure 5.

In Figure 5, the red graph represents the average capacity achieved using optimal condition cases described in (Equation 17). The green graph represents the achieved average capacity by use interim condition cases (e.g., $\phi =\frac{\pi }{8},\theta =\frac{3\pi }{8}$). The blue graph represents the most destructive average capacity value by placing the worst condition values (e.g., $\phi =\frac{\pi }{4}$, $\theta =\frac{\pi }{4}$). As we see in this section, the average capacity in all three scenarios in Figure 5 is on average less than ∼2 bit/s/Hz, compared to the performance of the three scenarios in Figure 3.

From an analysis of Figure 5, the conservation of two trends can be seen compared to the previous section. The first one is that there is a complete match between the simulator results and the theoretical analysis, i.e., the optimal values lead to maximum values of the channel capacity, and the fit is also proper concerning the intermediate conditions and the destructive conditions. The second one is that in both graphs, the red and the green, a significant increase occurs with the rise in values of $\frac{E_{b}}{N_{0}}$.

## 5. Analysis Performance-BER of OFDM-Dual-Polarization Combined with/without
Diversity-Techniques under MIMO-mmWave Channels with
Rain Distortions

The second part of this current study focuses on comparing the BER performance of a communication system without diversity techniques described in Section 3 to the performance of BER of a communication system with diversity techniques described in Section 4.

The communication system architecture that includes a combination of OSTBC-OFDM–MRRC techniques is very widespread, and many studies have been conducted around this topic [32,33,34,35,36]. Note that the integration of architecture of the communication system described in Section 4 is identical to the architecture in this section. The architecture included OFDM-OSTBC-MRRC, and XPI mechanisms, on the transmitter side and on the receiver side, which produce polarization-diversity capability. Note that in the communication model described in Section 2, the transmission matrix *S* has now become a combination of OSTBC-OFDM signals as provided in [32].

In both systems, there are implementations of the optimized XPI mechanism, and both systems are under rain distortion conditions. In this simulation, we focus on the BER performance by running three different scenarios (e.g., placing all the XPI values leading to optimal, mediocre, and poor channel conditions, identical to the process we did in Section 3 and Section 4) under the same rain distortion conditions. The block schemes of the simulator are similar to those described in Section 3 and Section 4.

In Figure 6, the BER-performance of the mmWave wireless communication system without diversity techniques under rain distortion can be seen. The three graphs represent different XPI values by tuning the XPI mechanism. The red graph represents the optimal XPI values, the green and the blue represent the mean XPI values, and the worst XPI values, respectively, are identical to the process in Section 3.

Figure 7 illustrates the BER-performance of the mmWave wireless communication system with OSTBC-MRRC and polarization diversity techniques under rain distortion. The three graphs also represent different XPI values identical to the process in Section 4.

The process of comparing performance generates three essential and significant insights. The most crucial insight is that by combining diversity techniques under a scenario of rain distortion in the DP-MIMO-mmWave channels, an improvement of ∼3 dB in the optimal graph (the red graph), and an improvement of ∼2 dB, in the mid graph (the green graph), compared to a system without a combination of techniques under the same conditions can be achieved.

The second insight is related to the compatibility between the results of the theoretical analysis of the three-channel conditions we analyzed and the results of the simulations. It means that in the graphical analysis of the channel performance and the visual examination of the BER performance, the three graphs obtained are ranked in terms of the simulation results following the theoretical analysis we performed.

The third insight is that the XPI values under certain conditions (for example, mistuning these mechanisms or setting specific values as analyzed in the theoretical process) may lead to the most destructive channel conditions and, in any case, lead to the worst performance - as reflected in the blue graph. This phenomenon appears regardless of the integration of diversity techniques in wireless communication systems or the lack of integration of these techniques.

## 6. Variation of Channel Matrix Components in the Transition between t to $t+t_{s}$ and Design of Orthogonal Conditions in the Channel Matrix with the Help of Feedback

Up to this point, our central assumption regarding the channel’s nature and statistical distribution was a Quasistatic Flat-Rayleigh–Fading. Despite the distortion in the transmitted signal polarization that occurs due to the rain phenomena, as we mentioned, it has been proven that with optimization and proper adjustment of the XPI mechanism, it is possible to achieve excellent performance in the leading parameters of the system.

Dealing with more stringent and challenging assumptions, for example, a selective channel in the time dimension produces a much more significant challenge. The combining MRRC rules are based on the assumption that the channel response components are consistent in a period of two-time symbol slots (e.g., for example, $h_{xx}\left(t\right)=h_{xx}(t+T_{s}$). Two essential constraints are added to this challenge of dealing with selective channel response. The first constraint is based on the fact that there is no physical possibility to create overhead information feedback or opportunity that is not limited in time so that feedback can share the Channel-State-Information-at-the-Transmitter (CSIT). (For example, in the scenario of selective channels in the spatial dimension. In this situation, the share of overhead information with the feedback is very limited). The second constraint is based on restricting the reevaluation time of the channel- response-matrix in each change of symbol time. That is, if the channel reevaluation time is higher than the symbol time, the decoding process in the receiver will ultimately collapse.

In this section, we offer a process of optimization that leads to solutions of efficient and fast feedback that deals with the phenomena of changes in multi-lane channels that change every symbol time.

In the first step, we assume the same communication model described in Section 2 (apart from the assumptions of $H_{TR}$, that we will describe in details below), with the same assumptions of rain distortion and architecture system that are described in Section 4. The XPI optimization is based on case 1 in Section 4.2. The second step, the optimization process is described as follows:

for $t_{0}=t$ the channel matrix response, $H_{TR}$, is:(18)$$\begin{array}{cc} H_{TR}\left(t_{0}\right) & =\left[\begin{array}{cc} h_{x,x}\left(t_{0}\right) & h_{x,y}\left(t_{0}\right)\\ h_{y,x}\left(t_{0}\right) & h_{y,y}\left(t_{0}\right) \end{array}\right], \end{array}$$

and for $t_{1}=t+T_{S}$, the channel matrix response, $H_{TR}$, is:(19)$$\begin{array}{cc} H_{TR}\left(t_{1}\right) & =\left[\begin{array}{cc} h_{x,x}\left(t_{1}\right) & h_{x,y}\left(t_{1}\right)\\ h_{y,x}\left(t_{1}\right) & h_{y,y}\left(t_{1}\right) \end{array}\right]. \end{array}$$

Applying the pre and post polarization operations and gathering the $t_{0}$ and $t_{1}$ channel matrices, we get:(20)$$\begin{array}{cc} \; & H_{TR}\left(t_{0},t_{1}\right)=\\ \; & =\left[\begin{array}{cc} Ah_{xy}\left(t_{0}\right)+Bh_{xx}\left(t_{0}\right)+Ch_{yx}\left(t_{0}\right)+Dh_{yy}\left(t_{0}\right) & Ah_{xx}\left(t_{0}\right)+Bh_{xy}\left(t_{0}\right)+Ch_{yy}\left(t_{0}\right)+Dh_{yx}\left(t_{0}\right)\\ Ah_{yy}\left(t_{0}\right)+Bh_{yx}\left(t_{0}\right)+Ch_{xx}\left(t_{0}\right)+Dh_{xy}\left(t_{0}\right) & Ah_{yx}\left(t_{0}\right)+Bh_{yy}\left(t_{0}\right)+Ch_{xy}\left(t_{0}\right)+Dh_{xx}\left(t_{0}\right)\\ Ah_{xy}\left(t_{1}\right)+Bh_{xx}\left(t_{1}\right)+Ch_{yx}\left(t_{1}\right)+Dh_{yy}\left(t_{1}\right) & Ah_{xx}\left(t_{1}\right)+Bh_{xy}\left(t_{1}\right)+Ch_{yy}\left(t_{1}\right)+Dh_{yx}\left(t_{1}\right)\\ Ah_{yy}\left(t_{1}\right)+Bh_{yx}\left(t_{1}\right)+Ch_{xx}\left(t_{1}\right)+Dh_{xy}\left(t_{1}\right) & Ah_{yx}\left(t_{1}\right)+Bh_{yy}\left(t_{1}\right)+Ch_{xy}\left(t_{1}\right)+Dh_{xx}\left(t_{1}\right) \end{array}\right]. \end{array}$$
Now, the columns of $H_{TR}\left(t_{0},t_{1}\right)$ are orthogonal if and only if
$$\begin{array}{cc} & \left(Ah_{x,y}{\left(t_{0}\right)}^{*}+Bh_{x,x}{\left(t_{0}\right)}^{*}+Ch_{y,x}{\left(t_{0}\right)}^{*}+Dh_{y,y}{\left(t_{0}\right)}^{*}\right)\cdot \\ & \cdot \left(Ah_{x,x}\left(t_{0}\right)+Bh_{x,y}\left(t_{0}\right)+Ch_{y,y}\left(t_{0}\right)+Dh_{y,x}\left(t_{0}\right)\right)+\\ & +\left(Ah_{y,y}{\left(t_{0}\right)}^{*}+Bh_{y,x}{\left(t_{0}\right)}^{*}+Ch_{x,x}{\left(t_{0}\right)}^{*}+Dh_{x,y}{\left(t_{0}\right)}^{*}\right)\cdot \\ & \cdot \left(Ah_{y,x}\left(t_{0}\right)+Bh_{y,y}\left(t_{0}\right)+Ch_{x,y}\left(t_{0}\right)+Dh_{x,x}\left(t_{0}\right)\right)+\\ & +\left(Ah_{x,y}{\left(t_{1}\right)}^{*}+Bh_{x,x}{\left(t_{1}\right)}^{*}+Ch_{y,x}{\left(t_{1}\right)}^{*}+Dh_{y,y}{\left(t_{1}\right)}^{*}\right)\cdot \\ & \cdot \left(Ah_{x,x}\left(t_{1}\right)+Bh_{x,y}\left(t_{1}\right)+Ch_{y,y}\left(t_{1}\right)+Dh_{y,x}\left(t_{1}\right)\right)+\\ & +\left(Ah_{y,y}{\left(t_{1}\right)}^{*}+Bh_{y,x}{\left(t_{1}\right)}^{*}+Ch_{x,x}{\left(t_{1}\right)}^{*}+Dh_{x,y}{\left(t_{1}\right)}^{*}\right)\cdot \\ & \cdot \left(Ah_{y,x}\left(t_{1}\right)+Bh_{y,y}\left(t_{1}\right)+Ch_{x,y}\left(t_{1}\right)+Dh_{x,x}\left(t_{1}\right)\right)=0, \end{array}$$
which is equivalent to
$$\begin{array}{cc} & \left(h_{x,y}\left(t_{0}\right)h_{x,x}{\left(t_{0}\right)}^{*}+h_{x,y}\left(t_{1}\right)h_{x,x}{\left(t_{1}\right)}^{*}+h_{y,y}\left(t_{0}\right)h_{y,x}{\left(t_{0}\right)}^{*}+h_{y,y}\left(t_{1}\right)h_{y,x}{\left(t_{1}\right)}^{*}\right)A^{2}+\\ & +\left(\begin{array}{c} h_{x,x}\left(t_{0}\right)h_{x,x}{\left(t_{0}\right)}^{*}+h_{x,x}\left(t_{1}\right)h_{x,x}{\left(t_{1}\right)}^{*}+\\ +h_{x,y}\left(t_{0}\right)h_{x,y}{\left(t_{0}\right)}^{*}+h_{x,y}\left(t_{1}\right)h_{x,y}{\left(t_{1}\right)}^{*}+\\ +h_{y,x}\left(t_{0}\right)h_{y,x}{\left(t_{0}\right)}^{*}+h_{y,x}\left(t_{1}\right)h_{y,x}{\left(t_{1}\right)}^{*}+\\ +h_{y,y}\left(t_{0}\right)h_{y,y}{\left(t_{0}\right)}^{*}+h_{y,y}\left(t_{1}\right)h_{y,y}{\left(t_{1}\right)}^{*} \end{array}\right)AB+\\ & +\left(\begin{array}{c} h_{x,x}\left(t_{0}\right)h_{y,x}{\left(t_{0}\right)}^{*}+h_{y,x}\left(t_{0}\right)h_{x,x}{\left(t_{0}\right)}^{*}+\\ +h_{x,x}\left(t_{1}\right)h_{y,x}{\left(t_{1}\right)}^{*}+h_{y,x}\left(t_{1}\right)h_{x,x}{\left(t_{1}\right)}^{*}+\\ +h_{x,y}\left(t_{0}\right)h_{y,y}{\left(t_{0}\right)}^{*}+h_{y,y}\left(t_{0}\right)h_{x,y}{\left(t_{0}\right)}^{*}+\\ +h_{x,y}\left(t_{1}\right)h_{y,y}{\left(t_{1}\right)}^{*}+h_{y,y}\left(t_{1}\right)h_{x,y}{\left(t_{1}\right)}^{*} \end{array}\right)AC+\\ & +\left(2h_{x,y}\left(t_{0}\right)h_{y,x}{\left(t_{0}\right)}^{*}+2h_{y,y}\left(t_{0}\right)h_{x,x}{\left(t_{0}\right)}^{*}+2h_{x,y}\left(t_{1}\right)h_{y,x}{\left(t_{1}\right)}^{*}+2h_{y,y}\left(t_{1}\right)h_{x,x}{\left(t_{1}\right)}^{*}\right)AD+\\ & +\left(h_{x,x}\left(t_{0}\right)h_{x,y}{\left(t_{0}\right)}^{*}+h_{x,x}\left(t_{1}\right)h_{x,y}{\left(t_{1}\right)}^{*}+h_{y,x}\left(t_{0}\right)h_{y,y}{\left(t_{0}\right)}^{*}+h_{y,x}\left(t_{1}\right)h_{y,y}{\left(t_{1}\right)}^{*}\right)B^{2}+\\ & +\left(2h_{x,x}\left(t_{0}\right)h_{y,y}{\left(t_{0}\right)}^{*}+2h_{y,x}\left(t_{0}\right)h_{x,y}{\left(t_{0}\right)}^{*}+2h_{x,x}\left(t_{1}\right)h_{y,y}{\left(t_{1}\right)}^{*}+2h_{y,x}\left(t_{1}\right)h_{x,y}{\left(t_{1}\right)}^{*}\right)BC+\\ & +\left(\begin{array}{c} h_{x,x}\left(t_{0}\right)h_{y,x}{\left(t_{0}\right)}^{*}+h_{y,x}\left(t_{0}\right)h_{x,x}{\left(t_{0}\right)}^{*}+\\ +h_{x,x}\left(t_{1}\right)h_{y,x}{\left(t_{1}\right)}^{*}+h_{y,x}\left(t_{1}\right)h_{x,x}{\left(t_{1}\right)}^{*}+\\ +h_{x,y}\left(t_{0}\right)h_{y,y}{\left(t_{0}\right)}^{*}+h_{y,y}\left(t_{0}\right)h_{x,y}{\left(t_{0}\right)}^{*}+\\ +h_{x,y}\left(t_{1}\right)h_{y,y}{\left(t_{1}\right)}^{*}+h_{y,y}\left(t_{1}\right)h_{x,y}{\left(t_{1}\right)}^{*} \end{array}\right)DB+\\ & +\left(h_{x,x}\left(t_{0}\right)h_{x,y}{\left(t_{0}\right)}^{*}+h_{x,x}\left(t_{1}\right)h_{x,y}{\left(t_{1}\right)}^{*}+h_{y,x}\left(t_{0}\right)h_{y,y}{\left(t_{0}\right)}^{*}+h_{y,x}\left(t_{1}\right)h_{y,y}{\left(t_{1}\right)}^{*}\right)C^{2}+\\ & +\left(\begin{array}{c} h_{x,x}\left(t_{0}\right)h_{x,x}{\left(t_{0}\right)}^{*}+h_{x,x}\left(t_{1}\right)h_{x,x}{\left(t_{1}\right)}^{*}+\\ +h_{x,y}\left(t_{0}\right)h_{x,y}{\left(t_{0}\right)}^{*}+h_{x,y}\left(t_{1}\right)h_{x,y}{\left(t_{1}\right)}^{*}+\\ +h_{y,x}\left(t_{0}\right)h_{y,x}{\left(t_{0}\right)}^{*}+h_{y,x}\left(t_{1}\right)h_{y,x}{\left(t_{1}\right)}^{*}+\\ +h_{y,y}\left(t_{0}\right)h_{y,y}{\left(t_{0}\right)}^{*}+h_{y,y}\left(t_{1}\right)h_{y,y}{\left(t_{1}\right)}^{*} \end{array}\right)CD+\\ & +\left(h_{x,y}\left(t_{0}\right)h_{x,x}{\left(t_{0}\right)}^{*}+h_{x,y}\left(t_{1}\right)h_{x,x}{\left(t_{1}\right)}^{*}+h_{y,y}\left(t_{0}\right)h_{y,x}{\left(t_{0}\right)}^{*}+h_{y,y}\left(t_{1}\right)h_{y,x}{\left(t_{1}\right)}^{*}\right)D^{2}=0, \end{array}$$
which is also equivalent to
(21)$$\begin{array}{cc} \; & \left(h_{x,y}\left(t_{0}\right)h_{x,x}{\left(t_{0}\right)}^{*}+h_{x,y}\left(t_{1}\right)h_{x,x}{\left(t_{1}\right)}^{*}+h_{y,y}\left(t_{0}\right)h_{y,x}{\left(t_{0}\right)}^{*}+h_{y,y}\left(t_{1}\right)h_{y,x}{\left(t_{1}\right)}^{*}\right)\left(A^{2}+D^{2}\right)+\\ \; & +\left(\begin{array}{c} {\left|h_{x,x}\left(t_{0}\right)\right|}^{2}+{\left|h_{x,x}\left(t_{1}\right)\right|}^{2}+\\ +{\left|h_{x,y}\left(t_{0}\right)\right|}^{2}+{\left|h_{x,y}\left(t_{1}\right)\right|}^{2}+\\ +{\left|h_{y,x}\left(t_{0}\right)\right|}^{2}+{\left|h_{y,x}\left(t_{1}\right)\right|}^{2}+\\ +{\left|h_{y,y}\left(t_{0}\right)\right|}^{2}+{\left|h_{y,y}\left(t_{1}\right)\right|}^{2} \end{array}\right)\left(AB+CD\right)+\\ \; & +\left(\begin{array}{c} h_{x,x}\left(t_{0}\right)h_{y,x}{\left(t_{0}\right)}^{*}+h_{y,x}\left(t_{0}\right)h_{x,x}{\left(t_{0}\right)}^{*}+\\ +h_{x,x}\left(t_{1}\right)h_{y,x}{\left(t_{1}\right)}^{*}+h_{y,x}\left(t_{1}\right)h_{x,x}{\left(t_{1}\right)}^{*}+\\ +h_{x,y}\left(t_{0}\right)h_{y,y}{\left(t_{0}\right)}^{*}+h_{y,y}\left(t_{0}\right)h_{x,y}{\left(t_{0}\right)}^{*}+\\ +h_{x,y}\left(t_{1}\right)h_{y,y}{\left(t_{1}\right)}^{*}+h_{y,y}\left(t_{1}\right)h_{x,y}{\left(t_{1}\right)}^{*} \end{array}\right)\left(AC+DB\right)+\\ \; & +2\left(h_{x,y}\left(t_{0}\right)h_{y,x}{\left(t_{0}\right)}^{*}+h_{y,y}\left(t_{0}\right)h_{x,x}{\left(t_{0}\right)}^{*}+h_{x,y}\left(t_{1}\right)h_{y,x}{\left(t_{1}\right)}^{*}+h_{y,y}\left(t_{1}\right)h_{x,x}{\left(t_{1}\right)}^{*}\right)AD+\\ \; & +\left(h_{x,x}\left(t_{0}\right)h_{x,y}{\left(t_{0}\right)}^{*}+h_{x,x}\left(t_{1}\right)h_{x,y}{\left(t_{1}\right)}^{*}+h_{y,x}\left(t_{0}\right)h_{y,y}{\left(t_{0}\right)}^{*}+h_{y,x}\left(t_{1}\right)h_{y,y}{\left(t_{1}\right)}^{*}\right)\left(B^{2}+C^{2}\right)+\\ \; & +2\left(h_{x,x}\left(t_{0}\right)h_{y,y}{\left(t_{0}\right)}^{*}+h_{y,x}\left(t_{0}\right)h_{x,y}{\left(t_{0}\right)}^{*}+h_{x,x}\left(t_{1}\right)h_{y,y}{\left(t_{1}\right)}^{*}+h_{y,x}\left(t_{1}\right)h_{x,y}{\left(t_{1}\right)}^{*}\right)BC=0. \end{array}$$

Let
$$\left\{\begin{array}{c} a=h_{x,y}\left(t_{0}\right)h_{x,x}{\left(t_{0}\right)}^{*}+h_{x,y}\left(t_{1}\right)h_{x,x}{\left(t_{1}\right)}^{*}+h_{y,y}\left(t_{0}\right)h_{y,x}{\left(t_{0}\right)}^{*}+h_{y,y}\left(t_{1}\right)h_{y,x}{\left(t_{1}\right)}^{*}\\ b=h_{x,y}\left(t_{0}\right)h_{y,x}{\left(t_{0}\right)}^{*}+h_{y,y}\left(t_{0}\right)h_{x,x}{\left(t_{0}\right)}^{*}+h_{x,y}\left(t_{1}\right)h_{y,x}{\left(t_{1}\right)}^{*}+h_{y,y}\left(t_{1}\right)h_{x,x}{\left(t_{1}\right)}^{*}\\ c=h_{x,x}\left(t_{0}\right)h_{y,x}{\left(t_{0}\right)}^{*}+h_{y,x}\left(t_{0}\right)h_{x,x}{\left(t_{0}\right)}^{*}+h_{x,x}\left(t_{1}\right)h_{y,x}{\left(t_{1}\right)}^{*}+h_{y,x}\left(t_{1}\right)h_{x,x}{\left(t_{1}\right)}^{*}+\\ +h_{x,y}\left(t_{0}\right)h_{y,y}{\left(t_{0}\right)}^{*}+h_{y,y}\left(t_{0}\right)h_{x,y}{\left(t_{0}\right)}^{*}+h_{x,y}\left(t_{1}\right)h_{y,y}{\left(t_{1}\right)}^{*}+h_{y,y}\left(t_{1}\right)h_{x,y}{\left(t_{1}\right)}^{*}\\ d={\left|h_{x,x}\left(t_{0}\right)\right|}^{2}+{\left|h_{x,x}\left(t_{1}\right)\right|}^{2}+{\left|h_{x,y}\left(t_{0}\right)\right|}^{2}+{\left|h_{x,y}\left(t_{1}\right)\right|}^{2}+\\ +{\left|h_{y,x}\left(t_{0}\right)\right|}^{2}+{\left|h_{y,x}\left(t_{1}\right)\right|}^{2}+{\left|h_{y,y}\left(t_{0}\right)\right|}^{2}+{\left|h_{y,y}\left(t_{1}\right)\right|}^{2}. \end{array}\right.$$

Then, (Equation 21) is equivalent to
(22)$$a\left(A^{2}+D^{2}\right)+d\left(AB+CD\right)+c\left(AC+DB\right)+2bAD+a^{*}\left(B^{2}+C^{2}\right)+2b^{*}BC=0.$$

Now, since $trace\left(H_{TR}{\left(t_{0},t_{1}\right)}^{*}H_{TR}\left(t_{0},t_{1}\right)\right)$ is a function of the form of *f* from Lemma 2, it follows that it is maximized at one of the exposed points (Equation 17). Therefore, in view of (Equation 22), over the points of maximum, the columns of $H_{TR}\left(t_{0},t_{1}\right)$ are orthogonal if and only if (a=0 and equivalently $a^{*}=0$):(23)$$h_{x,y}\left(t_{0}\right)h_{x,x}{\left(t_{0}\right)}^{*}+h_{y,y}\left(t_{0}\right)h_{y,x}{\left(t_{0}\right)}^{*}+h_{x,y}\left(t_{1}\right)h_{x,x}{\left(t_{1}\right)}^{*}+h_{y,y}\left(t_{1}\right)h_{y,x}{\left(t_{1}\right)}^{*}=0.$$

The third step is an application of conditions with the help of the proposed effective feedback and combining this feedback between the transmitter-receiver according to the architecture described in Section 4. The results of the simulation on the MATLAB/SIMULINK platform in terms of BER and average capacity are depicted in the illustrations Figure 8 and Figure 9, respectively.

In this simulation, we create a MIMO channel response engine that changes the channel values in each time slot of an icon. The capacity and BER calculators measure BER performance and capacity performance throughout the duration of the simulation. The last step is to activate the feedback that implements the condition (Equation 23).

In Figure 8, a comparison was carried out between two graphs representing the BER performance of the two scenarios we have specified. The blue graph represents the BER performance of the communication system under the destructive effect of changing the MIMO response channel in every symbol time slot, without any feedback facility. The red graph represents the BER performance under the same scenario with combining of feedback.

Beyond the distinction in improving performance in the system in which we incorporated the proposed feedback, two significant insights can be discerned. The first insight is that the system, without any feedback in the low region of the SNR, starts with a high error threshold and drags this threshold (that is not capable of producing information) to a value of almost ∼20 dB. (Remind that the decoding threshold value in terms of BER value, in such a case, is ∼10${}^{-3}$).

The second insight is a difference of almost ∼10 dB, about a given BER value of ∼10${}^{-4}$, a figure that indicates significant achievement of diversity gain with the feedback, and this is in addition to the fact that the feedback copes very well with the destructive effects of changes in the channel between the symbol times as we described.

Figure 9 represents the average capacity under the two scenarios above. As we can see, the capacity performance with the feedback is not significantly better than without the feedback, although the BER performance is much better with the feedback than without it. The reason for this is as follows. While the capacity depends on the the sum of the eigenvalues of $H_{TR}H_{TR}^{*}$ that can be the same for MIMO with spatial-diversity or without spatial-diversity, the BER depends on the spatial-diversity and on the orthogonality of the paths. Note that when the spatial-diversity is slight, even if the capacity is high, the system performance is unstable, since any slight distortion can cause a total collapse. We therefore recommend to use the feedback, even when the capacity is the dominant issue, because it decreases significantly the channel uncertainty, as can be seen from the BER performance.

## 7. Conclusions

In this paper, we have considered mmWaves wireless communication systems in two different DP architectures. We dealt with rain distortion effects and selective channels, caused by changes in the channel-response-matrix, during each time symbol duration.

In addition, we have formulated a proper communication model that accounts for all the mechanisms of the communication system. Regarding the proposed model, we have synthesized optimal conditions, in terms of the XPI mechanism, in order to achieve optimal capacity and minimal BER.

We have illustrated the need for intelligent use of diversity techniques in light of the above-mentioned destructive phenomena. Although diversity techniques in general improve significantly MIMO channel performance, under rain distortions they degrade significantly the channel capacity and BER performance. In view of this conflicting state of affairs, and in view of the common denominator of MIMO channels with or without diversity techniques, occupied by XPI mechanisms, we have found it as a trade-off mechanism between capacity and BER that can be used to improve significantly MIMO channel performance in the above-mentioned scenarios.

We have introduced the attractiveness of using smart feedback that optimizes the BER by the preservation of the orthogonality of the channel paths, and robustly improves and stabilizes the capacity performance.

We have demonstrated the motivation to use smart and fast feedback, creating a feedback development path that makes OSTBC techniques unnecessary in terms of wasting information and covering various situations in the channel. We have shown that using the proposed feedback scheme increases the speed of data transfer while maintaining channel-path orthogonality and achieving high diversity-gain.

In our future research, we would like to develop these feedback schemes and demonstrate their capacity to cope with selective channels in time and frequency domains. We would like to develop and integrate machine-learning techniques with our feedback schemes, in order to deal with destructive effects in the medium that include disruptive offsets while optimizing conflicting system performance keys.

## Figures and Tables

**Figure 1 sensors-22-07182-f001:**
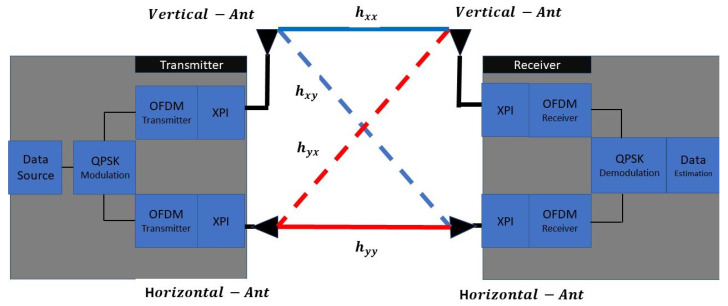
Dual Polarization-MIMO Scheme.

**Figure 2 sensors-22-07182-f002:**
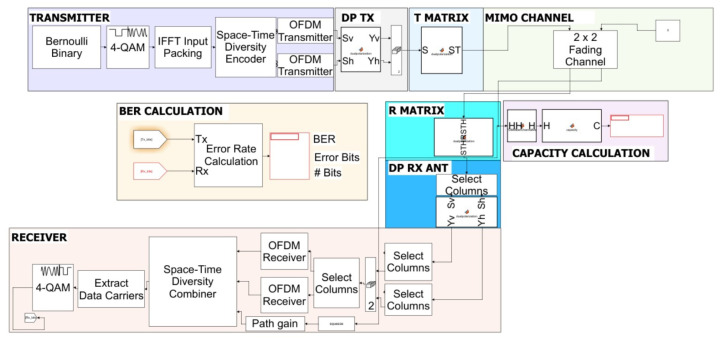
Block Diagram of The Simulator in The SIMULINK/MATLAB Platform.

**Figure 3 sensors-22-07182-f003:**
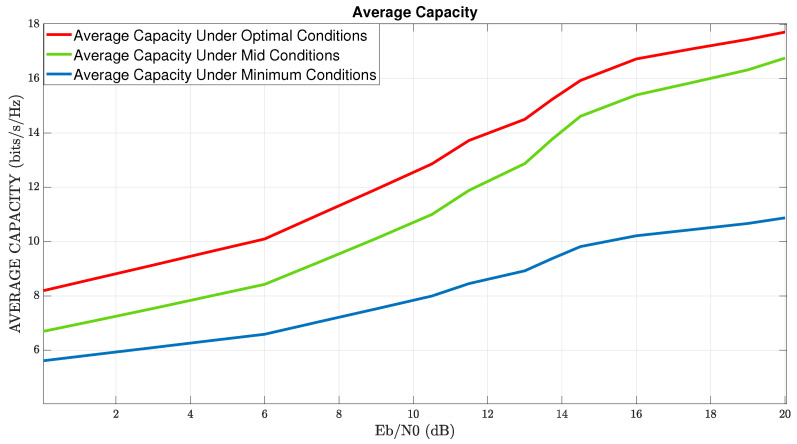
Average Capacity Of Channel MIMO With Dual-Polarization And XPI Power Allocation.

**Figure 4 sensors-22-07182-f004:**
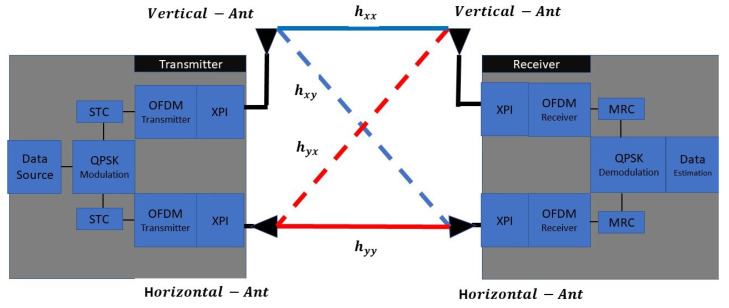
Dual Polarization-OSTBC-MIMO Scheme.

**Figure 5 sensors-22-07182-f005:**
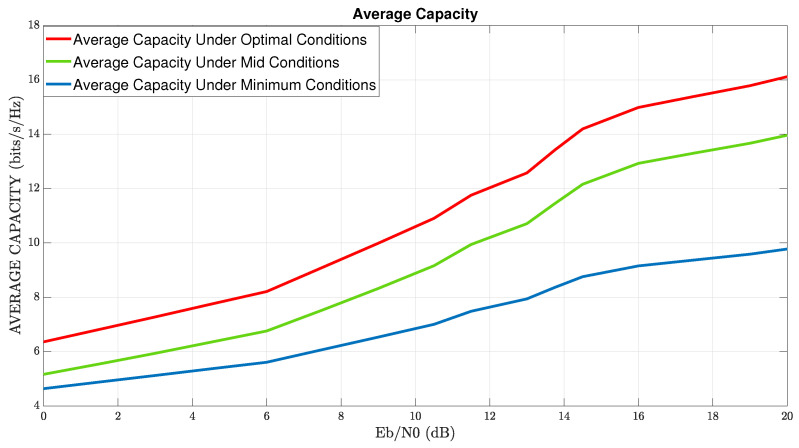
Average Capacity Under Optimal, Mid, and Minimum Conditions with Diversity-Techniques.

**Figure 6 sensors-22-07182-f006:**
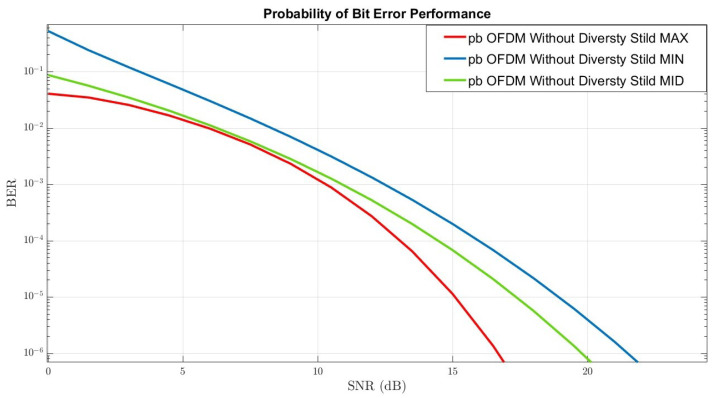
BER Performance of OSTBC-OFDM-MRRC Without Diversity-Techniques.

**Figure 7 sensors-22-07182-f007:**
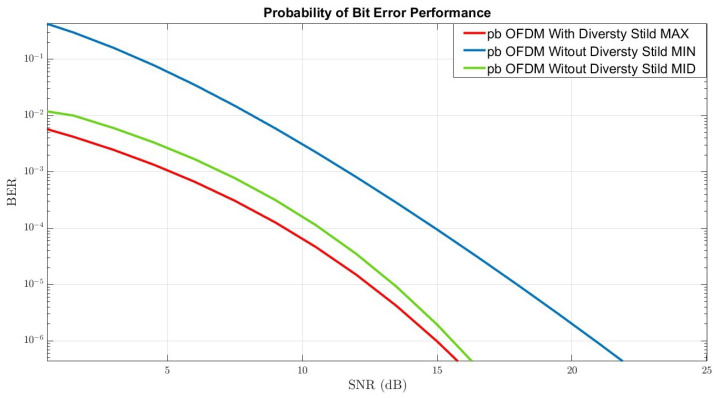
BER Performance of OSTBC-OFDM-MRRC With Diversity-Techniques.

**Figure 8 sensors-22-07182-f008:**
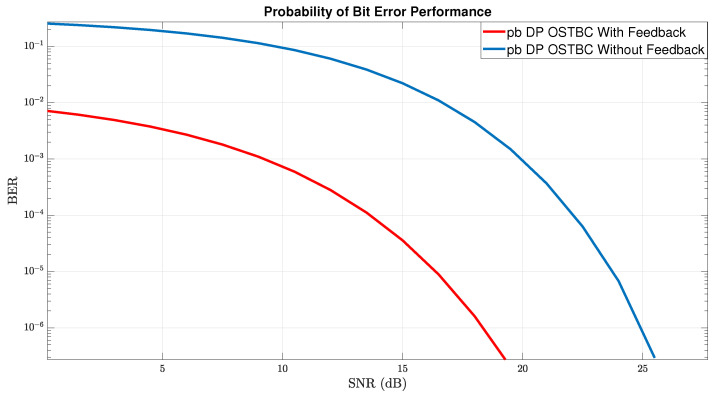
BER OSTBC With/Without a Feedback.

**Figure 9 sensors-22-07182-f009:**
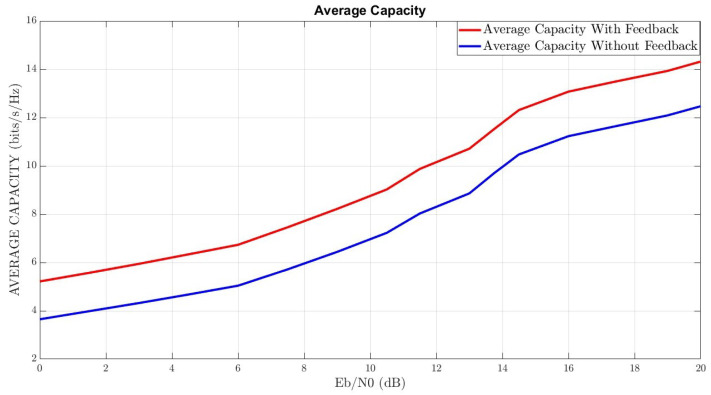
Average Capacity With/Without Feedback.

**Table 1 sensors-22-07182-t001:** Definition of four optimal channels.

	θ=0	θ=π/2
ϕ=0	$H_{{}_{TR}}=\left(\sqrt{\frac{1}{1+k}}\right)\cdot \left[\begin{array}{cc} h_{xx} & h_{xx}\delta_{x,y}\\ h_{xx}\delta_{y,x} & h_{yy} \end{array}\right]$	$H_{{}_{TR}}=\left(\sqrt{\frac{1}{1+k}}\right)\cdot \left[\begin{array}{cc} h_{yy}\delta_{y,x} & h_{yy}\\ h_{xx} & h_{xx}\delta_{x,y} \end{array}\right]$
ϕ=π/2	$H_{{}_{TR}}=\left(\sqrt{\frac{1}{1+k}}\right)\cdot \left[\begin{array}{cc} h_{xx}\delta_{x,y} & h_{xx}\\ h_{yy} & h_{yy}\delta_{y,x} \end{array}\right]$	$H_{{}_{TR}}=\left(\sqrt{\frac{1}{1+k}}\right)\cdot \left[\begin{array}{cc} h_{yy} & h_{yy}\delta_{y,x}\\ h_{xx}\delta_{x,y} & h_{xx} \end{array}\right]$

**Table 2 sensors-22-07182-t002:** Units and Parameters used in the conducted simulations.

Symbol	Parameter	Value/Description
$c_{q}$	Number of OSTBC component encoders in the transmitter	q=1,2
$N_{TX}$	Total number of transmission antennas in the MGSTC transmitter	2
$N_{RX}$	Sum of the total receiver antennas in the Receiver	2
$\begin{array}{c} S_{c}=\left[\begin{array}{c} \vec{s_{c,1}}\\ \vec{s_{c,2}} \end{array}\right] \end{array}$	Row vectors of block transmission symbol matrix	$\vec{s_{c,1}},\vec{s_{c,2}}$
	Antenna Design MATLAB Object	Phased.shortDipole AntennaElement
	Simulation running time	10 [ms]
	Frequency operation	k-band 18–27 [GHz]
	Duplex mode	TDD
$T_{S_{c}}$	Sample time per frame	16 [μs]
	Modulation	QPSK
$R_{b}$	Transmission bit rate	23.5 [Mbps]
SNR	Signal-to-Noise Ratio	0 to 25 [dB]
	Received power	−80 to −90 [dBm]
$E_{s}$	Energy to symbol	1 [Watt]
	FFT length	64
BW	Bandwidth	6 [MHz]
	Rain rate intensity range	0 to 400 [mm/h]

**Table 3 sensors-22-07182-t003:** Definition of complex channel gain between the transmit and receive antennas.

	${\mathrm{RX}}_{1}$	${\mathrm{RX}}_{2}$
$TX_{1}$	$h_{11}$	$h_{12}$
$TX_{2}$	$h_{21}$	$h_{22}$

**Table 4 sensors-22-07182-t004:** Transmission OSTBC coding operation.

	*t*	$t+T_{s}$
$TX_{1}$	$s_{1}$	$-s_{2}^{*}$
$TX_{2}$	$s_{2}$	$s_{1}^{*}$

**Table 5 sensors-22-07182-t005:** Received signal *Y* and additive white Gaussian noise (AWGN) signal *Z*.

	*t*	$t+T_{s}$
$YX_{1}$	$Y_{1}\left(1\right)$ $Z_{1}\left(1\right)$	$Y_{1}\left(2\right)$ $Z_{1}\left(2\right)$
$YX_{2}$	$Y_{2}\left(1\right)$ $Z_{2}\left(1\right)$	$Y_{2}\left(2\right)$ $Z_{2}\left(2\right)$

## Data Availability

Not applicable.

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
