# Peer review of "Analysis of OSTBC-OFDM Combined with Dual-Polarization and Time-Diversity in Millimeter-Wave MIMO Channels with Rain Distortions"

_sensors, 2022, doi:10.3390/s22197182_

Round 1

Reviewer 1 Report

The aim of the study is about the optimal use of combining Orthogonal-Space-Time-Coding (OSTBC) and Maximal-Ratio-Receive-Combiner (MRRC) with cross-polarization diversity techniques. This type of study is a new field in sensors which is appropriate to publish in this journal.

Author Response

15/09/2022
Dear Dr. Ms. Hedy Ji
Section Managing Editor

We are writing you in regard with our paper entitled:
"Analysis of OSTBC-OFDM Combined with Dual-Polarization and Time-Diversity in Millimeter- Wave MIMO Channels with Rain Distortions" (Manuscript ID sensors-1926032).
We first wish to thank you for finding and convening qualified reviewers to examine our paper. We would also like to thank you very much for your professional and quick handling of our article.
We thank the reviewers for the encouraging and professional responses that help us to improve our article.
We accepted all the comments of the respected reviewers and fix all the comments point by point in accordance of their remarks.
We believe that the readers of the MDPI-Sensors-Smart Antennas for Future Communications will be benefited from the elaborate study.

Sincerely,

The authors:

RF Engineer Mr. Avner Elgam,

Dr. for Mathematics and Computer Sciences Yossi Peretz,

Prof. for Communication Systems Yosef Pinhasi.

Reviewer 1:

English language and style are fine/minor spell check required.

Answer:
Following this remark, we have reviewed the English throughout the article. The new manuscript is an improved version—corrections were introduced in those places that were required.

Reviewer 2 Report

The article is trying to formulate a wireless communication MIMO model inmillimeter wave (mmWave) channels under the rain distortion.This article demonstrate optimal conditions for combining Orthogonal Space-Time Coding (OSTBC) and Maximum Ratio Receive Combiner (MRRC) using XPI tuning using cross-polarization diversity techniques.Furthermore, the authors propose a procedure that approximates closed-loop MIMO to achieve a global maximum channel capacity while maintaining channel path orthogonality to minimize BER.The research content is original and clearly defined, and has practical significance in the field. And the research content is within the scope of the journal.

I think the author should use a diagram to illustrate the destructive phenomenon in the introduction to increase the readability of the article. Appropriate reductions are made in Section 4 to highlight optimized performance. Authors should consider simplifying the results section, without repeating the research methods of the paper. Improve the clarity of the picture in Figure 5.

Author Response

15/09/2022
Dear Dr. Ms. Hedy Ji
Section Managing Editor

We are writing you in regard with our paper entitled:
"Analysis of OSTBC-OFDM Combined with Dual-Polarization and Time-Diversity in Millimeter- Wave MIMO Channels with Rain Distortions" (Manuscript ID sensors-1926032).
We first wish to thank you for finding and convening qualified reviewers to examine our paper. We would also like to thank you very much for your professional and quick handling of our article.
We thank the reviewers for the encouraging and professional responses that help us to improve our article.
We accepted all the comments of the respected reviewers and fix all the comments point by point in accordance with their remarks.
We believe that the readers of the MDPI-Sensors-Smart Antennas for Future Communications will be benefited from the elaborate study.

Sincerely,

The authors:

RF Engineer Mr. Avner Elgam,

Dr. for Mathematics and Computer Sciences Yossi Peretz,

Prof. for Communication Systems Yosef Pinhasi.

Reviewer: 2

  1. I think the author should use a diagram to illustrate the destructive phenomenon in the introduction to increase the readability of the article.

Answer:

Thank you very much for your comment. In the introduction (line 50), a reference to a significant source discusses the demonstration of the destructive phenomena we are dealing with in this article.
In the source we cited, the authors discuss the issues of the effect of rain on the millimeter wave communication channel and present the destructive phenomena in figure 2. e. 
In addition, we have added a reference to a source that presents a diagram illustrating the destructive phenomena we are dealing with in the current article.

2. Authors should consider simplifying the results section, without repeating the research methods of the paper.

Answer:

Thanks so much for your response. Following your comment, we added refinement in the introduction regarding the description of the article's structure and what topic each chapter deals with.
On the one hand, we wanted to maintain a reading sequence in the eyes of the reader, so we briefly mentioned a short introduction explaining the technique we present in the chapter at the beginning of each chapter.
On the other hand, we emphasized the uniqueness that each chapter presents and the advantages and disadvantages of the differences between the various techniques presented in the different chapters.

3. Improve the clarity of the picture in Figure 5.

Answer:

In the new version we fixed all the fonts and the clarity of all the figures, especially figure 5.

Reviewer 3 Report

In general a well written paper with solid mathematical formulation. I have some minor comments prior publication

1) Very few details are provided on the overall simulation setup. Please justify the selected parameters and possibly insert a table.

2) The quality of some figures can be improved, such as fig. 5

Author Response

15/09/2022
Dear Dr. Ms. Hedy Ji
Section Managing Editor

We are writing you in regard to our paper entitled:
"Analysis of OSTBC-OFDM Combined with Dual-Polarization and Time-Diversity in Millimeter- Wave MIMO Channels with Rain Distortions" (Manuscript ID sensors-1926032).
We first wish to thank you for finding and convening qualified reviewers to examine our paper. We would also like to thank you very much for your professional and quick handling of our article.
We thank the reviewers for the encouraging and professional responses that help us to improve our article.
We accepted all the comments of the respected reviewers and fix all the comments point by point in accordance with their remarks.
We believe that the readers of the MDPI-Sensors-Smart Antennas for Future Communications will be benefited from the elaborate study.

Sincerely,

The authors:

RF Engineer Mr. Avner Elgam,

Dr. for Mathematics and Computer Sciences Yossi Peretz,

Prof. for Communication Systems Yosef Pinhasi.

Reviewer: 3

1. Very few details are provided on the overall simulation setup. Please justify the selected parameters and possibly insert a table.

Answer:

Thanks so much for your response. Note that the phenomena we deal with and that we present in our article do occur in real life based on the critical work of the following articles we cited:

Huang, J.; Cao, Y.; Raimundo, X.; Cheema, A.; Salous, S. Rain statistics investigation and 623
rain attenuation modeling for millimeter wave short-range fixed links. IEEE Access 2019, 624
7, 156110–156120.

Torkildson, E.; Zhang, H.; Madhow, U. Channel modeling for millimeter wave MIMO. In 604
Proceedings of the 2010 Information Theory and Applications Workshop (ITA). IEEE, 2010, pp. 605
1–8

Based on the leading parameters in which these phenomena occur, we established the simulator and developed the simulator with great accuracy of these parameters. The justification for choosing these parameters is reflected in many articles we cited in the introduction and in the rest of our article, especially in the two articles we mentioned.
Matlab/Simulink platform enables accurate simulations that describe these phenomena based on modeling all the mechanisms that make up the communication system we described.
Indeed, we have received your critical response and have compiled and added to the article a table of parameters on which the construction of the article and its results are based.

2.The quality of some figures can be improved, such as fig. 5.

Answer:

In the new version we fixed all the fonts and the clarity of all the figures, especially figure 5.